# Isolation and Cultivation of Adipose-Derived Mesenchymal Stem Cells Originating from the Infrapatellar Fat Pad Differentiated with Blood Products: Method and Protocol

**DOI:** 10.3390/mps6010003

**Published:** 2022-12-25

**Authors:** Markus Neubauer, Karina Kramer, Johannes Neugebauer, Lukas Moser, Anna Moser, Dietmar Dammerer, Stefan Nehrer

**Affiliations:** 1Center for Regenerative Medicine and Orthopedics, Danube University Krems, Dr. Karl-Dorrek-Str. 30, 3500 Krems, Austria; 2Department of Orthopedics and Traumatology, Karl Landsteiner University of Health Sciences, University Hospital Krems, Mitterweg 10, 3500 Krems, Austria

**Keywords:** mesenchymal-stem-cells, adipose-tissue, infra-patellar-fat pad, blood products, joint preservation, regenerative orthopedics

## Abstract

Adipose-derived mesenchymal stem cells (ASCs) are a promising source for clinical application in regenerative orthopedics. ASCs derived from the infra-patellar fat pad (IFP)—a distinct adipose structure in the knee—show superior regenerative potential compared to subcutaneous-fat-derived cells. Furthermore, it has been shown that blood products enhance ASCs’ viability. A major challenge for clinical translation of both ASCs and blood products is the low comparability of obtained data due to non-standardized harvesting, isolation and preparation methods. The aim of this method-paper is to provide reproducible protocols to help standardize basic research in the field to build a sound basis for clinical translation with an emphasize on practicability. The presented protocols include (i) ASC isolation from the IFP, (ii) blood product preparation and (iii) ASC incubation with blood products.

## 1. Introduction

A standardized protocol for (surgical) harvest and laboratory processing of adipose-derived mesenchymal stem cells (ASCs) is critical to ensure comparability of data. Sound basic data are key to foster translation. The aim of this method paper is to provide a protocol for ASCs’ harvest and basic processing with regard to easy applicability.

Mesenchymal stem cells were shown to be beneficial in orthopedic conditions such as osteoarthritis (OA) and cartilage regeneration amongst others [1,2]. Their medicinal effects are explained both by the modulating effects of their secretome as well as their differentiation potential into various cell types, such as chondrocytes [1,3]. In parallel, clinical evidence for MSCs therapy is increasing [4].

Adipose tissue is an increasingly used source for clinical use of MSCs. The abundance of material and the low comorbidity are primary reasons for this trend.

The common way of harvesting ASCs from the subcutaneous tissue is via liposuction. This technique is usually unfamiliar to orthopedic surgeons and associated with a moderate risk [5].

Infra-patellar-fat-pad-derived ASCs (IFPs) are an alternative. This fat pad is an intracapsular structure that lies extrasynovially in the knee joint. IFPs were shown to have superior chondrogenic potential compared to subcutaneous-fat-derived MSCs [6,7,8]. 

The surgical reduction of this fat pad often is an obligatory step in open and arthroscopic knee surgery. It has been shown to be safe and helps reduce pain associated with fat pad inflammation [9,10,11]. These tissue remnants are considered surgical waste and discarded.

Since the first clinical trials with ASCs, a commonly used strategy in the clinic to foster ASC therapeutic features was to combine them with blood products [12,13]. The rationale for this combination was to support ASC growth and support viability [14,15,16].

Two main areas of IFPs use in a “point-of-care” setting in the operating room seem reasonable: for (early onset) OA and for tissue engineering on scaffolds. A routine use of IFPs in combination with autologous blood products during open knee surgeries or arthroscopy has the potential to improve standard orthopedic care.

Approaching this goal requires scientific data starting with basic investigations of how harvesting and isolation processes alter IFPs to help design optimized clinical trials. A major challenge in the acquisition of those data is represented by the in-homogenous protocols and methods that make comparability difficult. The presented protocol was shown to be easily applicable in former projects and is still in use in ongoing projects at the Center for Regenerative Medicine Krems.

The aim of this paper is to meet this challenge by providing easily repeatable protocols for IFPs’ harvest and basic processing.

## 2. Procedure and Experimental Design

The following subsection describe protocols used for each step of experimental designs that use IFP-MSCs. In most cases, step-by-step experimental instructions as well as the time needed per step.

### 2.1. Surgical Cell Harvest

The surgical harvest of the adipose tissue from the infra-patellar fat pad is performed either during open joint surgeries or arthroscopically.

In the case of open joint surgery, such as total-knee arthroplasty, the majority of the infra-patellar fat pad is removed with the rationale to avoid painful impingement. In this case, a bigger tissue part is gathered. For reasons of practicability, it is recommended to ask an assistant to mechanically mince the fat pad with a scalpel in the surgical room under sterile conditions and send those fragments for further processing to the lab. Fragmenting the tissue in this way further lowers the risk of infection both of cells in the incubator as well as for technicians in the lab that are usually less familiar with using scalpels.

In the case of arthroscopic harvest, the surgeon has to focus on resecting the fat pad only without any other tissues such as synovial membrane, cartilage or plicae. Usually, a standard shaver system is used to resect the IFP. The integrated sucking system transports the minced IFP tissue parts out of the joint. The surgeon then uses a sterile sieve to collect the tissue. Moreover, sieves that are integrated into the sucking system are commercially available. However, often they do get clogged quickly which is why the authors did use an extra, sterile sieve to collect the tissue (Figure 1). As a surgical side note, the IFP should not be totally resected to avoid Hoffa-post-resection syndrome.

The adipose tissue is then stored in a PBS and antibiotics solution at +4°. The temperature must be monitored and documented both for storage and transport to guarantee stable conditions and avoid influences of deviating temperatures. The transport is organized in a cryo-box with a validated USB temperature tracker that helps to obtain an instantaneous read-out before acceptance in the lab. Tissue should be transported to the lab in a matter of 24 h.

### 2.2. Ethical Votes

Usually, the Hoffa fat pad or parts of it are classified as “surgical waste”. In consequence some institutions claim no necessity for informed consent or ethical approval. Dragoo et al. stated in a paper from 2017 that “…institutional review board categorized the specimens as surgical waste products not requiring separate written consent…” [17]. Nevertheless, the authors recommend obtaining both an ethical approval and an informed consent form from each patient donating cells to avoid incidences regarding tissue ownership alongside other legal and ethical concerns. Likewise, it is advised to obtain a positive ethical vote for drawing blood in order to produce blood products for experimental use (current project at the Center for Regenerative Medicine: (i) Ethics Committee of the Evangelic hospital Vienna (Positive Vote: 3 May 2018) and (ii) Commission for Scientific Integrity and Ethics Karl Landsteiner University Krems EC Nr: 1022/2020).

### 2.3. Cell Isolation and Cell Culture

The fragmented tissue is further processed in the lab. A flow chart for the isolation steps are presented in Figure 2.

First, it is weighed and documented. Thereafter, the tissue is enzymatically digested by adding collagenase solution (catalogue number: C0130, Sigma-Aldrich, St. Louis, MO, USA): 9000 units of collagenase I/15 mL DMEM/10 g of fat. The reaction tube afterwards is put in the incubator for 2 h at 37 °C in a shaking machine (Figure 3).

The next step is the filtration, which is conducted by typically using a 40 μm Cell Strainer (Corning^®^ 40 μm Cell Strainer, catalogue number: 431,750, Durham, NC, USA). Neutralization is performed with a standard MSCs growing media (DMEM, high glucose, GlutaMAXTM Supplement, pyruvate, catalogue number: 31,966,047, Gibco Life Technologies Europe Bv, Bleiswijk, The Netherlands) and addition of antibiotics. The typical ingredients our research group uses are 2% Penicillin/Streptomycin (catalogue number: P4333, Sigma-Aldrich Chemie GmbH, Steinheim, Germany) and 1% Amphotericin B (catalogue number: A2942, Sigma-Aldrich Chemie GmbH, Steinheim, Germany), 10% fetal calf serum (catalogue number: 11,550,356, GibcoTM, qualified, heat inactivated, Gibco Life Technologies Europe Bv, Bleiswijk, The Netherlands), 1% non-essential amino acids (MEM NEAA catalogue number: 11,140,050, Gibco Life Technologies Europe Bv, Bleiswijk, The Netherlands) and bFGF 1 ng/mL (Fibroblast Growth Factor-Basic, human, catalogue number: F0291, Sigma-Aldrich Chemie GmbH, Steinheim, Germany).

The following step is a centrifugation at 700× *g* for 10 min. Thereafter, the supernatant is removed. The remaining cells are then ready for seeding aiming for a density of 10,000 cells/cm^2^ in standard MSC growth media. The desired confluency should be approximately around 80–90%. Nonadherent cells are discarded the first time after 24 h and then every 2–3 days (Figure 4).

The detachment process is enzymatically induced by accutase (catalogue number: SCR005, Sigma-Aldrich Chemie GmbH, Steinheim, Germany). The resulting solution is again centrifuged in the above-mentioned manner.

After detachment, the experimenter either intends to split the cells or to freeze them for later use. 

In the case of a decision for freezing, dilution of cells deviates from the above-mentioned seeding protocol. In this case, media that protect cells from damage by freezing are chosen with 10% DMSO (catalogue number: D8418, Sigma-Aldrich Chemie GmbH, Steinheim, Germany) and 90% FCS (catalogue number: 11,550,356, Gibco^TM^). The targeted cell count was 1 million cells per 1 mL of freezing media. Thereafter, cells are cryopreserved in liquid nitrogen. However, the freezing is a two-step procedure. This solution is transferred to cryovials and stored in alcohol-free cell freezing containers at −80 °C freezer overnight. The further transport to the liquid nitrogen tank can be performed the next day.

The thawing in order to re-use cells is another sensitive step to use them for further experiments. Samples should be transferred immediately from liquid nitrogen to a 37 °C water bath for less than 1 min. Afterwards, thawed samples are diluted in regular standard and previously heated MSC growing media and centrifuged. It is advisable to routinely check cells viability after thawing. In the protocol of the center >70% of viable cells is considered desirable for further use. In this case, a density of 10,000 cells/cm^2^ in standard MSCs growing media is typically targeted.

Figure 5 shows Flow Cytometry results with 3 MSC-specific surface markers (CD90, CD105, CD73) to demonstrate successful MSC isolation. Please find the details of how to apply the Flow cytometric analysis in Section 2.5.

### 2.4. Blood Product Preparation Protocol

Platelet-rich plasma (PRP): Whole blood is drawn from donors’ standard blood vials, such as 9 mL VACUETTE vials, containing an anticoagulant of either K3EDTA (catalogue number: 454,217; Greiner Bio-One, Kremsmünster, Austria) or Trinatriumcitrat (catalogue number: 455,322; Greiner Bio-One, Kremsmünster, Austria). The key element of PRP is the centrifugation protocol. At the Center for Regenerative Medicine, a simple protocol with 2 centrifugation steps is carried out to gain leucocyte-poor PRP: 1. First centrifugation of the whole blood at 440× *g* for 10 min at room temperature right after blood drawing. Thereby, 3 layers are formed in the test-tube: (i) a bottom layer—predominantly consistent of red blood cells, (ii) a middle layer—containing the so-called buffy coat and (iii) top layer—predominantly plasma. 2. Aspiration of the plasma layer. Please be careful not to contaminate your aspiration with the buffy coat. This aspirate is transferred into a separate, usually 15 mL standard tube (without any ingredients). 3. Second centrifugation of the aspirate at 1700× *g* for 10 min. 4. Resuspension of the formed platelet pellet in 50% of the remaining (platelet-poor) supernatant.

The blood products obtained by this protocol can either be directly used for experiments of can be stored at −80 °C for later use.

### 2.5. Flow Cytometry

According to a consensus statement from the International Society for Cellular Therapy in 2006 (ISCT), cells must show a minimum the below listed features to be defined as “(human) mesenchymal stem cells” [18]. 

Plastic adherent (in standard culture);Expression of CD73, CD90 and CD105; Absence of CD34, CD45, CD14, HLA-DR, CD11b and CD19Chondrogenic, osteogenic and adipogenic differentiation potential.

The protocol the authors’ use consists of positive marker for CD73, CD105, CD90. In contrast, the following surface markers should not be expressed: CD34, CD11b, CD19, CD45, HLA-DR. 

This meets the standard minimum definition criteria besides plastic adherence and differentiation.

Cells should be investigated by flow cytometry directly after isolation with passage P0. However, in the case of very limited cell yields or other unfavorable pre-requisites, P1 cells are feasible to be checked for CD markers as well.

The exact steps are as follows: IFPs are cultured to 90% of confluency in standard growth media. Routinely, media is changed every 3 days. Like mentioned above, cells are detached by applying Accutase centrifuged in a FACS buffer (1% (*v*/*v*) FCS in phosphate-buffered saline (PBS)) and stained thereafter by antibodies (the exact same products with the already-mentioned catalogue numbers are used as described above). Table 1 lists the antibodies used in our protocol (BD Stemflow hMSC Analysis Kit, BD Biosciences, St. Louis, MO, USA). For 30 min, samples are incubated with antibodies at room temperature and shielded from light. The flow is performed on live cells and negative gates are to be set on isotypecontrol.

(In the lab at Danube University a Flow cytometer FC500 is used (Beckman Coulter, Brea, CA, USA)).

### 2.6. Blood Product Supplemented Cell Incubation and Chondrogenic Cell Differentiation 

The chondrogenic differentiation stimulus requires special media. At the Center for Regenerative Medicine the following media composition is used: DMEM, high glucose, GlutaMAXTM Supplement (DMEM, high glucose, GlutaMAXTM Supplement, pyruvate, catalogue number: 31966047, Gibco Life Technologies Europe Bv, Bleiswijk, The Netherlands), pyruvate (catalogue number: 10569010), 1% ITS, 100 nM dexamethasone (catalogue number: D4902, Sigma-Aldrich Chemie GmbH, Steinheim, Germany), 50 μg/mL ascorbic acid (catalogue number: A4403, Sigma-Aldrich Chemie GmbH, Steinheim, Germany), 1% non-essential amino acids (MEM NEAA, catalogue number: 11140050, Gibco Life Technologies Europe Bv, Bleiswijk, The Netherlands), 5 ng/mL TGFbeta-3 (catalogue number: AF-100-36E, PeproTech, New York, NY, USA), 0.2 vol% methyl cellulose (Sigma-Aldrich Chemie GmbH, Steinheim, Germany).

In the case of blood product supplementation, usually 1% is added. To prevent clotting, addition of heparin of 2 U/mL is crucial. It was shown to be practical for experiments to form chondrogenic pellets and thereby create a 3D structure as opposed to a 2D one (Figure 6). To create chondrogenic differentiation pellets, 250,000 cells are placed in a 15 mL polypropylene tubes embedded in the above mentioned medium. Pellets are then again formed by at top speed of 4164× *g* for 10 min.

The usual differentiation period is 3 weeks. Change in media containing blood products is achieved firstly after 7 days and thereafter twice a week. We added a PCR example from 3 pooled IFP-MSC donors after chondrogenic differentiation, demonstrating successful differentiation with cartilage-specific gene expressions (ACAN, SOX9, COL2/COL1 correlation). Supplementation with 2 different blood products (PRP and HA—developed as a subsequent PRP) and fetal calf serum (FCS) was performed. FCS is included to show effects of this commonly used additive on MSCs’ differentiation. This graph is included to demonstrate successful differentiation only and does not intend to distract from the protocol by adding information about PCR analysis, statistics or any related matters. For further details, please find a description in our former publication where the same methods were applied [16] (Figure 7).

Additionally, histological samples stains with Alcian Blue and Hematoxylin Eosin are presented in Figure 8. Likewise, this graph is included to demonstrate successful differentiation only and does not intend to distract from the protocol by adding information about histological staining or any related matters. For further details, please find a description in our former publication where the same methods were applied [16].

## 3. Discussion

Investigating IFPs with standardized methods is a prerequisite to obtain sound, comparable data. However, the focus must be on translation and thus on practicability. The following discussion will address main issue with regard to (i) surgical and clinical aspects, (ii) biological aspects and (iii) legislative aspects due to their inter-relation for the translational process. 

An issue for orthopedic surgeons in the process of providing IFPs leads to unfavorable results for patients’ satisfaction. It is known that an aggressive resection of the Hoffa pad may lead to a “vacuum” phenome that retracts the patellar tendon and leads to a patella inferna [19]. However, Hoffa pad hypertrophy is associated with knee injuries and contributes to anterior knee pain [20,21]. Thus, Hoffa resection can be assumed to be generally a beneficial addition of the arthroscopy if it is not performed too aggressively.

The supplementation of MSCs with BPs is, as mentioned above, a common practice. The comparability of BPs due to a lack of standardization starting with the preparation and inter-donor variability adds another variable that must be mentioned as a limiting factor. This circumstance, however, also does imply the need for a commonly used protocol like the one presented. Moreover, fetal calf serum (FCS) is commonly used as a positive control group in differentiation experiments. This additive is very variable and poorly standardized. Moreover, FCS itself may have an impact upon the differentiation potential of MSCs. This is a limiting factor for any protocol that use FCS.

A biological issue may be the applicability of obtained lab data into clinical study designs as cells are exposed to further manipulation.

This concern may be met as follows: the aim of standardized protocols is to obtain comparable data for experiments such as cell viability after harvest. To research cells’ viability and similar cell features, likely related to beneficial clinical applications, lab processing is obligatory. However, laboratory “manipulation” by processing is obligatory but should be kept to a minimum. A way to achieve this aim is by using standardized protocols. The issue of passaging may be likewise addressed: as much as necessary to achieve sufficient cell counts to, for example, form 3D pellets but not more. However, recent studies recommend P3-5 even for clinical (re)use. Given the experience of the authors’ team, usually, P1-3 results in sufficient cells counts for further experiments, which may be a hint that this issue has to be noted but its’ effects are likely to be of minor relevance [22]. The meaning of “least manipulation” may be exemplified by the controversial definition “homologous use” versus “non-homologous use” cell use. In the U.S. is defined as “…the repair, reconstruction, replacement, or supplementation of a recipient’s cells or tissues with an HCT/P that performs the same basic function or functions in the recipient as in the donor (21 CFR 1271.3(c)), including when such cells or tissues are for autologous use…” [23]. In Europe, the definition is as follows: “…the repair, reconstruction, replacement, or supplementation of a recipient’s cells or tissues with an HCT/P that performs the same basic function or functions in the recipient as in the donor (21 CFR 1271.3(c)), including when such cells or tissues are for autologous use…” [24]. Therefore, adipose-derived stem cells from the subcutaneous tissue are not considered “homologous” when injected in joint cavities. These emerging questions require to be addressed systematically, starting with basic scientific investigations performed by using standardized protocols and methods that produce comparable data among research groups in order to foster translation. 

## Figures and Tables

**Figure 1 mps-06-00003-f001:**
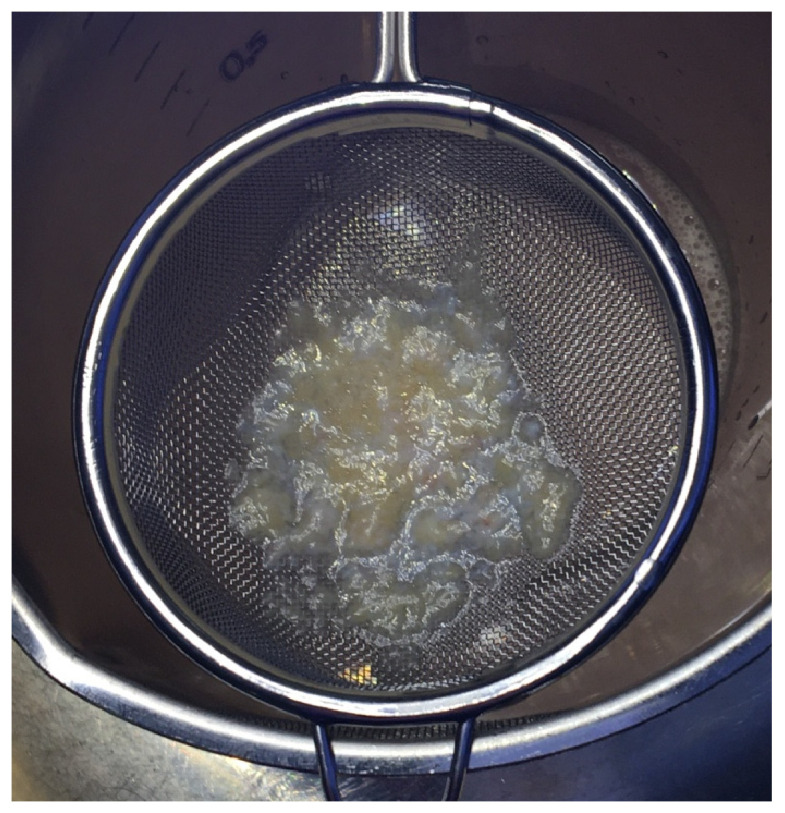
Arthroscopically harvested infra-patellar fat pad.

**Figure 2 mps-06-00003-f002:**
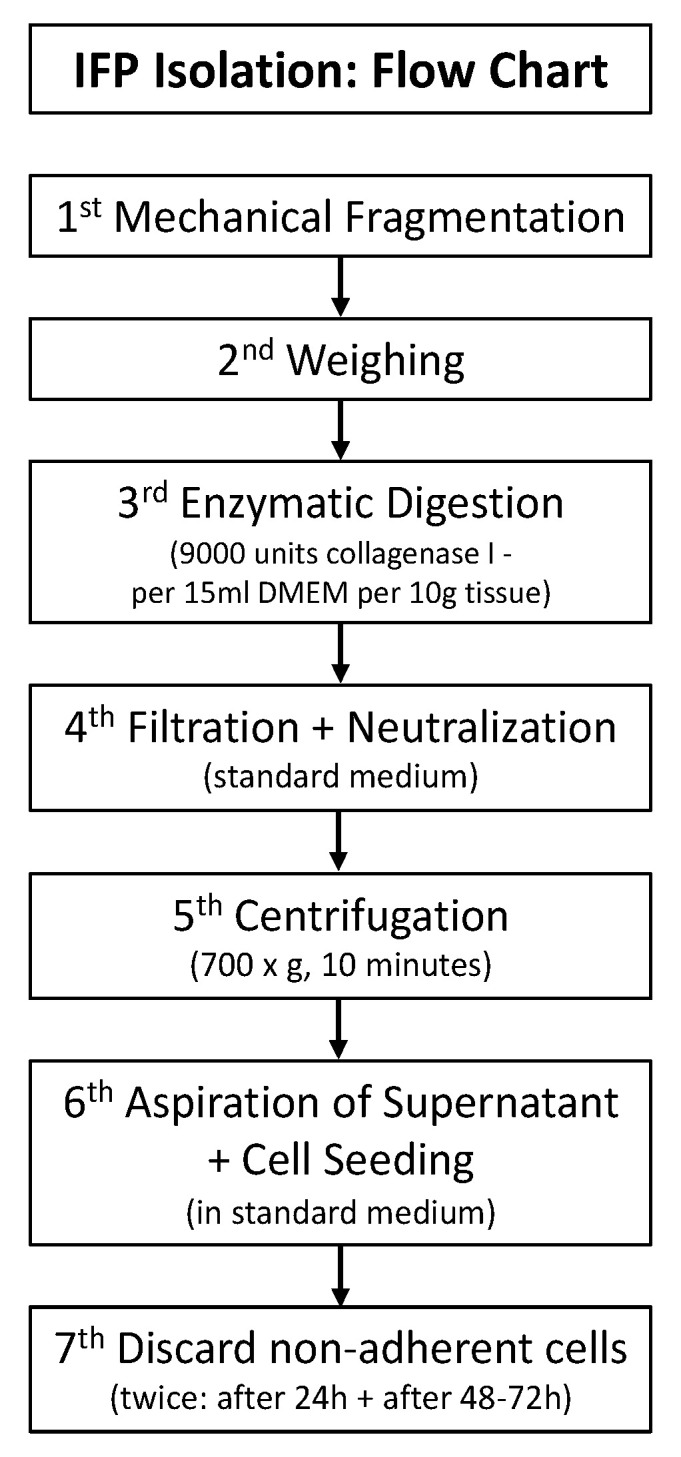
IFP isolation protocol.

**Figure 3 mps-06-00003-f003:**
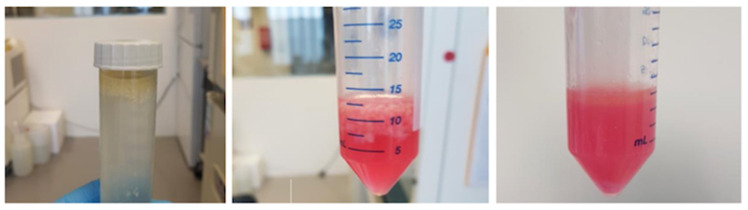
Processing of arthroscopically harvested IFP in the lab: (from left to right): (i) material from the operating room in PBS, (ii) material in medium before enzymatic digestion, (iii) material after enzymatic digestion.

**Figure 4 mps-06-00003-f004:**
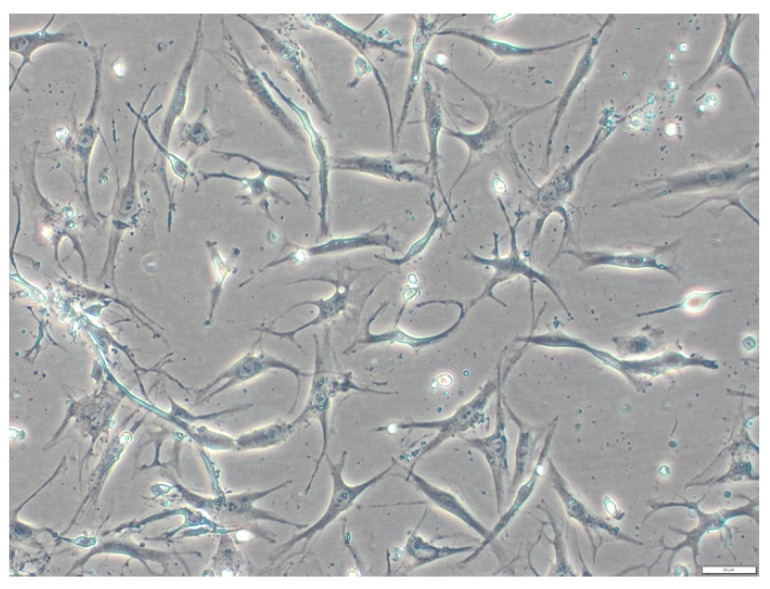
Cells after seeding: typical MSCs formation.

**Figure 5 mps-06-00003-f005:**
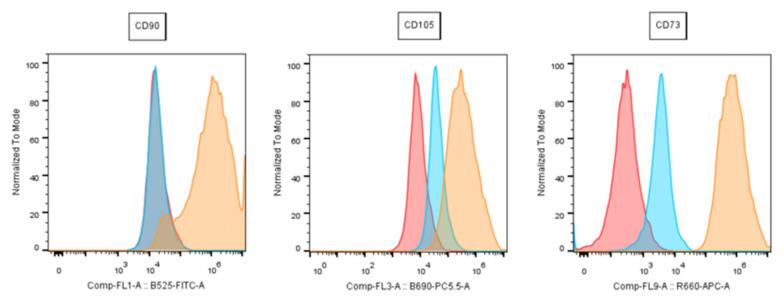
Flow Cytometry: results from a representative donor: red = cells only; orange = marker; blue = isotype; (CD = cluster of differentiation).

**Figure 6 mps-06-00003-f006:**
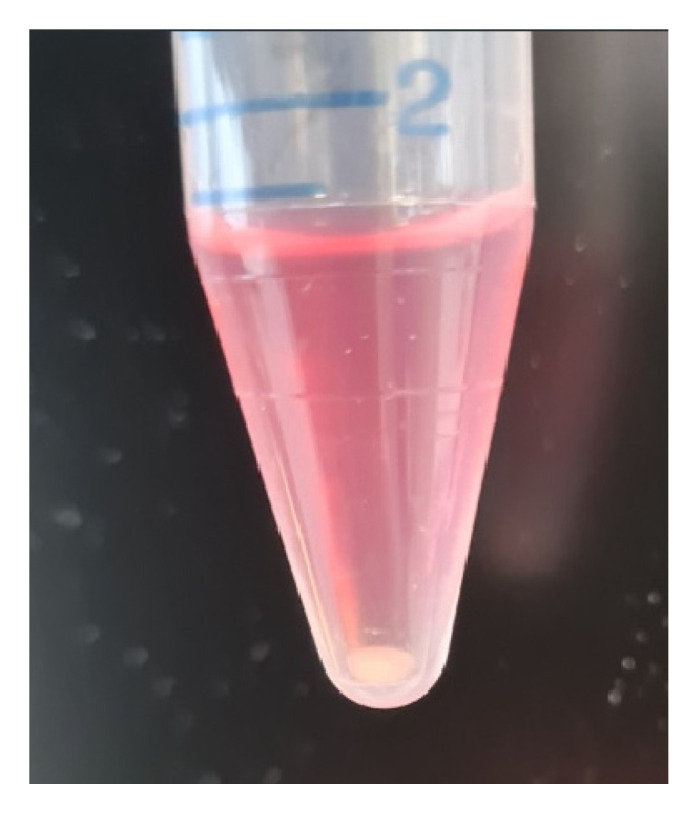
Chondrogenic pellet formation.

**Figure 7 mps-06-00003-f007:**
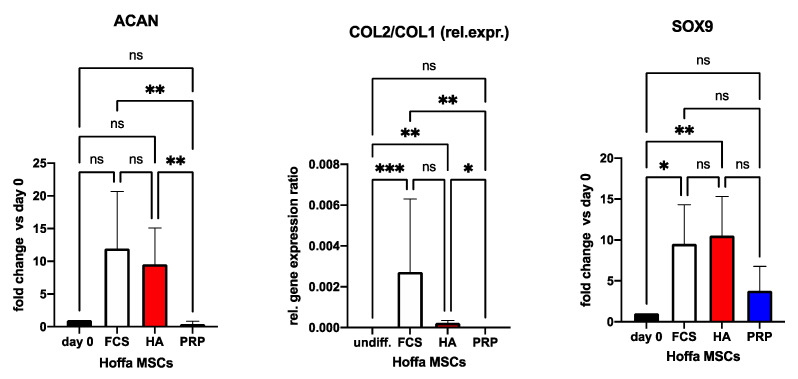
Gene expressions for chondrogenic differentiation with blood product (PRP or HA) supplementation. ACAN = aggrecan; COL1 = collagen 1A1 gene; COL2 = collagen 2 gene; PRP = platelet-rich plasma (blue); FCS = fetal calve serum (white); HA = hyperacute serum (red); SOX9 = factor 9. Results are expressed as the mean ± SD. In case of COL2/COL1 the relative expression is shown ∗ indicates a *p* < 0.05 as a significant difference in gene expression within one treatment group. ∗∗ represents *p* < 0.01 and ∗∗∗ represents *p* < 0.001. ns represents not significant.

**Figure 8 mps-06-00003-f008:**
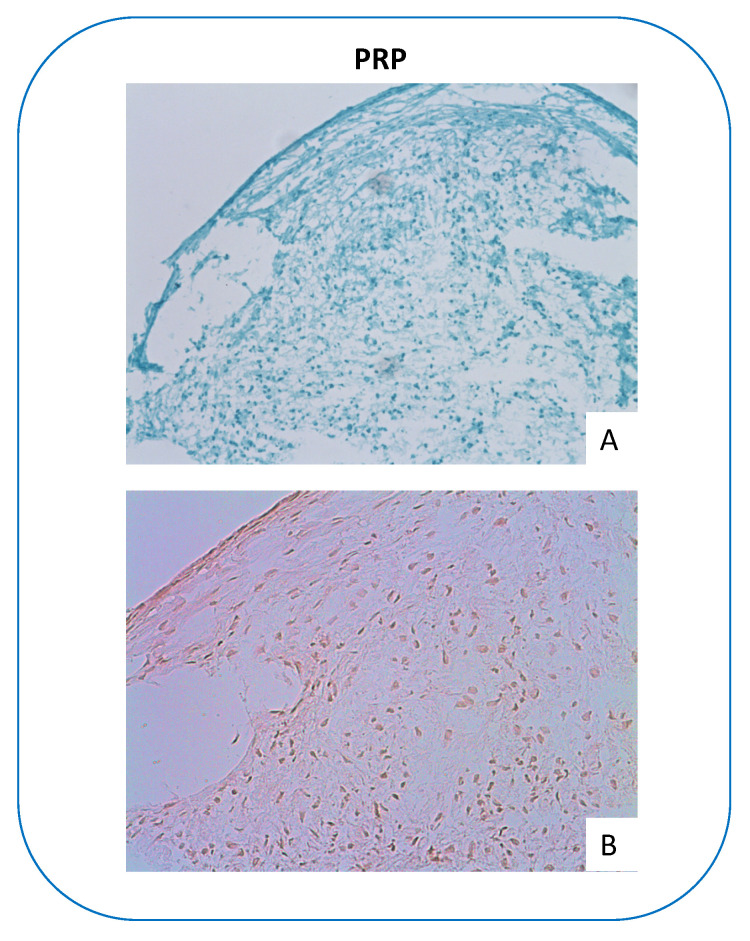
Histology results: PRP = platelet-rich-plasma; (**A**) = Alcian Blue staining, (**B**) = Hematoxylin Eosin staining.

**Table 1 mps-06-00003-t001:** Antibodies for Flow cytometric analysis.

Positive Markers	CD105 PerCP-Cy5.5/CD73 APC/CD90 FITC
**Additional Positive Drop-In Marker**	CD44 PE
**Negative Cocktail**	CD45/CD34/CD11b/CD19/HLA-DR
**Isotype Control**	mlgG1, κ PerCP-Cy5.5/mlgG1, κ APC/mlgG1, κ FITC (for positive markers) mIgG1, κ/mIgG2a, κ PE (for negative cocktail) mIgG2b, κ PE (for CD44 drop in)

## Data Availability

Data available on request due to restrictions, e.g., privacy or ethical. The data presented in this study are available on request from the corresponding author.

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
