# Peer review of "Isolation and Cultivation of Adipose-Derived Mesenchymal Stem Cells Originating from the Infrapatellar Fat Pad Differentiated with Blood Products: Method and Protocol"

_mps, 2022, doi:10.3390/mps6010003_

Round 1

Reviewer 1 Report

The manuscript addresses an important issue concerning non-standardized harvesting, isolation and preparation methods adipose-derived mesenchymal stem cells. In their manuscript the authors propose a reproducible protocol to help standardize basic research within regenerative medicine. Unfortunately, the manuscript does not provide clear evidence to indicate that the proposed protocol is more suitable for reproducibility as well as biological functionality of MSCs compared to other protocols from other laboratories. In this context, certain steps/points of the protocol contradict the goal of standardization or are not discussed adequately.

Introduction

Line 38: adipose-derived MSCs are not really novel

Line 50: the necessity of blood products “to bypass potential negative influences” is not clear; since addition (and preparation) represents another big variable in the procedure that should be standardized, a strong case for the necessity of such factors should be presented. Additionally, blood products will intrinsically carry donor-dependent variability.

Procedure/Experimental design

Line 82-85: Procedure not clear; please rewrite. Filter integrated into a shaver-system? Figure one would provide more value, if a schematic of the procedure was shown here.

Line 88: is there a specific reason for 8C? For the purpose of standardization 4C would be more suitable

Line 108: Figure 2 should be rearranged, bad use of space.

Line 125: Use of FCS represents one of the greatest challenges when attempting to establish standardized procedures. The batch-to-batch variation of this additive can have a huge effect on the differentiation potential and therefore on therapeutic characteristics of MSCs (regardless of the source) down the road. In this regard, even if the steps of the protocol are standardised and followed, use of FCS unfortunately renders it obsolete. This should at least be mentioned as a limitation.

Line 149 and 153: use of the word “ideally” and “recommended” should be avoided if standardization is the goal

Line 157: density of 10000 cells/cm2 seems high; usually it is around 4000-5000. Is there a specific reason for it?

Line 231: Figure 5. It is not sufficient to show a picture of a pellet in a 15ml falcon to claim successful chondrogenesis. At least histological assessment (alcian, picrosirius) and/or sGAG quantification with DMMB should be performed.

Line 252-256: not clear, please rephrase.

Line 258-259: These two sentences contradict each other; if standardization is key, alternative protocols should be avoided

Line 266: The regulatory section should be shortened and simplified. Although interesting, this is not the main topic of the manuscript and take considerable part of the discussion.

Reviewer 2 Report

Dear Authors, your manuscript aims to show a novel approach to isolate adipose stem cells ant how to use blood derivate supply to improve their differentiation. Even if the isolation procedure is well exposed, something is missing. I suggest you improve the manuscript with a characterization of the isolated cells in term of histological and genetic characteristics to demonstrate that you selected the type of cells you are interested in and that they do not express chondrogenic markers prior differentiation. Moreover, the differentiation part lacks data. There are no tests that demonstrate the chondrogenic differentiation of the cells. Please, add more information about this.

Round 2

Reviewer 1 Report

Thank you very much for submitting a revised version of the manuscript which considered all comments and suggestions adequately.

Reviewer 2 Report

Dear Authors, thank you very much for considering my suggestions. You integrated the text with valuable data that give strength to the research. In my opinion the manuscript can be published.